# Trajectories of frailty in aging: Prospective cohort study

Joe Verghese[1,2]*, Emmeline Ayers[1], Sanish Sathyan[1], Richard B. Lipton[1,3,4], Sofiya Milman[2,5], Nir Barzilai[2,5], Cuiling Wang[4]

1 Department of Neurology, Albert Einstein College of Medicine, Bronx, New York, United States of America,
2 Department of Medicine, Albert Einstein College of Medicine, Bronx, New York, United States of America,
3 Department of Psychiatry, Albert Einstein College of Medicine, Bronx, New York, United States of America,
4 Department of Epidemiology and Population Health, Albert Einstein College of Medicine, Bronx, New York, United States of America, 5 Department of Genetics, Albert Einstein College of Medicine, Bronx, New York, United States of America

* joe.verghese@einsteinmed.org

## Abstract

### Background

Emerging evidence suggests that there is significant variability in the progression of frailty in aging. We aimed to identify latent subpopulations of frailty trajectories, and examine their clinical and biological correlates.

### Methods

We characterized frailty using a 41-item cumulative deficit score at baseline and annual visits up to 12 years in 681 older adults (55% women, mean age 74·6 years). Clinical risk profile and walking while talking performance as a clinical marker of trajectories were examined. Mortality risk associated with trajectories was evaluated using Cox regression adjusted for established survival predictors, and reported as hazard ratios (HR). Proteome-wide analysis was done.

### Findings

Latent class modeling identified 4 distinct frailty trajectories: relatively stable (34·4%) as well as mild (36·1%), moderate (24·1%) and severely frail (5·4%). Four distinct classes of frailty trajectories were also shown in an independent sample of 515 older adults (60% women, 68% White, 26% Black). The stable group took a median of 31 months to accumulate one additional deficit compared to 20 months in the severely frail group. The worst trajectories were associated with modifiable risk factors such as low education, living alone, obesity, and physical inactivity as well as slower walking while talking speed. In the pooled sample, mild (HR 2·33, 95% CI 1·30–4·18), moderate (HR 2·49, 95% CI 1·33–4·66), and severely frail trajectories (HR 5·28, 95% CI 2·68–10·41) had higher mortality compared to the stable group. Proteomic analysis showed 11 proteins in lipid metabolism and growth factor pathways associated with frailty trajectories.

**Data Availability Statement:** The Institutional Review Board of the Albert Einstein College of Medicine has data access restrictions in place because health claims data are a sensitive data source, and have ethical restrictions imposed due

to concerns regarding privacy. Anonymized data are available to all interested individuals or institutions upon request, and completion of a data transfer agreement. Please contact the Einstein Institutional Review Board staff at (irb@einsteinmed.org) for all data requests.

**Funding:** The authors received no specific funding for this work. Parent study funding disclosed in manuscript.

**Competing interests:** The authors have declared that no competing interests exist.

## Conclusion

Frailty shows both stable and accelerated patterns in aging, which can be distinguished clinically and biologically.

## Introduction

Frailty is conceptualized as a state of decreased physiological reserve and compromised capacity to maintain homeostasis as a consequence of multiple, accumulated deficits in aging [1–3]. Frailty increases risk for numerous adverse outcomes including disability, falls, and death [2, 4]. Prevalence of frailty increases with age [1, 5], but studies of long-lived individuals indicate that many preserve physical functions even in extreme ages [6]. Many adults continue to have normal walking speeds, a major frailty criterion [1, 5], well into their ninth decade [6, 7]. This variation in frailty with aging suggests that there may be subgroups of individuals who differ in severity and rate of progression in frailty over time. Emerging evidence supports heterogeneity in frailty progression in aging with variations in individual rates of change [3, 8]. Previous studies using different definitions of frailty as well as methodological approaches have reported both stable and declining trajectories in older adults [8–12]. But these findings regarding different frailty courses were not cross-validated in independent samples, and none have explored the underlying biology of latent subpopulations of frailty trajectories [8, 12]. Identifying trends and trajectories is acknowledged as a high priority for frailty research [3, 8].

Our aims in this investigation were three-fold. First, to identify latent subpopulations of frailty trajectories that may differ in their rates of progression in a cohort of community-dwelling older adults, and validate these findings in an independent aging cohort. Second, to examine clinical risk profiles of the frailty trajectories as well as their association with mortality to establish their clinical relevance. We hypothesized that individuals with worse frailty trajectories would have unhealthier profiles and higher mortality risk. Third, understanding the biology of latent subpopulations of frailty trajectories may provide insights into new therapeutics for frailty. We reported a proteomic profile associated at cross-section with frailty with proteins related to lipid metabolism among the most significant [13]. Lipid metabolism has been implicated in aging and age related diseases [14]. Hence, we extended our cross-sectional finding to identify biological underpinnings of the longitudinal frailty trajectories. Understanding frailty trajectories and their clinical and biological correlates will help clinicians and patients in clinical prognostications, spur research into protective mechanisms against frailty, and help develop new interventions.

## Methods

### Study population

The LonGenity study, established in 2008 as a genetic discovery cohort, recruited Ashkenazi-Jewish adults age 65 and older, who were either offspring of parents with exceptional longevity (OPEL), defined as having at least one parent who lived to age 95 and older, or offspring of parents with usual survival (OPUS), defined as having neither parent who survived to age 95 [4, 15]. Participants were recruited using population lists or through community organizations and advertisements in local newspapers. Potential participants were contacted by telephone to assess eligibility, and invited for in-person evaluations. Exclusion criteria include cognitive impairment (score >8 on the Blessed-Information-Memory-Concentration test or >2 on the

AD8 dementia screen), severe visual loss, and having a sibling in the study [4, 15]. Cross-validation was done in the Einstein Aging Study (EAS), also based in our institution, a study of cognitive aging in community-dwelling non-demented adults age 70 and over in the Bronx [16, 17].

All participants signed written informed consents prior to enrollment. The Einstein institutional review board approved both study protocols, and conforms to the provisions of the Declaration of Helsinki.

Both cohorts received uniform clinical assessments at the same clinical site at baseline and annual visits [4, 7, 17]. We included participants with at least three frailty assessments to derive trajectories as done in prior studies [9]. Frailty assessments were implemented in LonGenity in 2008,[4] and 2004 in EAS [17]. Of the 1060 LonGenity participants seen from 2008 to 2020, we excluded 379 with less than 3 frailty assessments. The final sample for this analysis included 681 (64.2%) LonGenity participants. The median follow-up for the LonGenity cohort was 7.6 years (IQR 5·29–9·27). Between 2004 and 2015, 1072 non-demented individuals were assessed in EAS for frailty, and 515 (48.0%) EAS participants with 3 or more annual assessments were included in this analysis. The median follow-up for the EAS cohort was 5.1 years (IQR 3·08–7·43). Table 1 presents baseline characteristics of the two study samples included in this investigation. There were no significant statistical differences in demographics (age and sex) and baseline frailty status between the included and excluded participants in both the LonGenity and EAS cohorts.

## Frailty

The two common clinical methods to define frailty are as a cumulative deficit index or as a phenotype; with strengths and weaknesses noted for both [1–3, 5]. We used the frailty index (FI) as it is recommended for biological investigations, and is considered a robust marker of biological age [2, 18, 19]. Furthermore, compared to phenotypic definitions, FI provides a wider range of scores that may better capture the multidimensional and dynamic nature of

**Table 1. Baseline characteristics of LonGenity and Einstein Aging Study cohorts.**

| Variables | LonGenity | Einstein Aging Study |
|---|---|---|
| Source population, n | 1060 | 1072 |
| Study sample (≥3 waves), n (%) | 681 (64.2) | 515 (48.0) |
| Study years | 2008–2020 | 2004–2015 |
| Age years, mean ± SD | 74·6 ± 6·1 | 79·3 ± 5·1 |
| Women, % | 55·0 | 60·0 |
| Education years, mean ± SD | 17·8 ± 2·7 | 14·4 ± 3·4 |
| Race/ethnicity, % | | |
| White | 100·0 | 67·6 |
| Black | 0 | 25·6 |
| Hispanic | 0 | 5·4 |
| Other/unknown | 0 | 1·4 |
| Comorbidities score, mean ± SD | 1·2 ± 1·1 | 1·8 ± 2·0 |
| Walking speed, mean ± SD | 111·68 ± 19·02 (n = 554) | 98·19 ± 21·14 (n = 507) |
| Body mass index, mean ± SD | 27·4 ± 4·9 (n = 659) | 27·5 ± 4·8 (n = 475) |
| FCSRT, free recall, mean ± SD | 33·9 ± 4·9 (n = 678) | 31·8 ± 5·6 (n = 511) |
| Frailty Index score, mean ± SD | 0·1 ± 0·1 | 0·2 ± 0·1 |

SD: standard deviation, FCSRT: free and cued selective reminding test.

frailty over time [2, 3, 9]. We included 41 variables common to both cohorts, and collected using the same tests and questionnaires at the same center to derive the FI as previously described (S1 Table in S1 File) [4, 20]. Criteria to select FI variables were association with health status, biologically relevant, accumulates with age, not saturate at an earlier age, and represent multiple organ systems [21]. FI was calculated by adding number of deficits, and dividing the sum by number of variables per participant; resulting in a range of scores from 0 (no frailty) to 1 (complete frailty) [20]. All FI items are equally weighted reducing disproportionate effects from few variables. FI is comparable across studies even when different number or types of items are used. Phenotypic frailty was diagnosed using criteria proposed by Fried and colleagues [1, 17].

We used mixture or latent class trajectory modeling using SAS PROC TRAJ procedure to identify distinct trajectories of frailty [22], which allowed us to simultaneously estimate probabilities of subgroup membership and distinct trajectories within each subgroup. The analysis was done first in the LonGenity sample, and then independently repeated in the EAS sample. Maximum likelihood method was used to estimate parameters. It included participants with missing data based on missing at random assumption. Model selection and determination of number of subgroups was based on scientific plausibility, and Bayesian Information Criteria (BIC) to evaluate goodness of fit [23]. Given a fixed value k for the number of classes (denoted as K), we examined linear to cubic trends by testing of the parameters and model fitting statistics (BIC), to obtain the final model. Starting with k = 1, we followed the recommendation based on change in BIC between models with K = k+1 and K = k to determine whether additional class should be added [22]. Participants were assigned to the subgroup class with the highest estimated probability.

## Clinical profile

Self-reported clinical information was confirmed with medical records and informants when available. A 10-point comorbidities score was created by summing presence of the following illnesses:[4, 16] hypertension, myocardial infarction, heart failure, angina, diabetes, depression, chronic lung disease, stroke, Parkinson's disease, and arthritis. Cognitive status was characterized using Free and Cued Selective Reminding (memory) and Digit Symbol Substitution tests (non-memory) [4]. Body mass index (BMI) was calculated as weight in kilograms divided by square of height in meters.

Participants in both cohorts were tested with the same instrumented walkway (GAITRite®, NJ) while walking at normal pace and while reciting alternate letters of the alphabet (walking while talking test: WWT) [7, 17]. The WWT is a real-world test of divided attention, and predicted frailty in EAS [17]. We examined walking speed (cm/s) during WWT at baseline as a clinical marker of frailty trajectories. Mobility-related questions accounted for only 2 (4·8%) out of the 41 FI items, but the FI did not include questions about ability to walk while talking or divide attention. Death was ascertained from designated contacts,[15] and supplemented by National Death Index searches [15].

## Proteomics

The proteome is the complete set of proteins that can be expressed by an organism at a given time [13]. Proteomic analysis in LonGenity has been described [13, 24]. In brief, plasma samples were analyzed using 5·0k SomaScan assay platform (SomaLogic Inc., Boulder, Co) that includes 5209 SOMAmer reagents that recognize human proteins. After standardization, 4265 SOMAmer reagents were available for analysis [13, 24]. Relative fluorescence unit values

observed after data normalization procedures for each SOMAmer reagent were natural log transformed. Outliers were removed using median absolute deviation method [13, 24].

## Statistical analysis

Baseline characteristics across trajectory classes were compared by ANOVA. Logistic regression adjusted for age and sex was used to compare baseline WWT speed categories between stable and mild frail versus moderate and severely frail classes in LonGenity, and reported as odds ratio (OR) with 95% confidence intervals. Threshold of significance was 0.05. Cox models were used to compare mortality risk between trajectories (stable reference) in both cohorts separately, and then pooled, and reported as hazard ratios (HR) adjusted for age, sex, years of education and comorbidities score. LonGenity analysis was adjusted for OPUS-OPEL status, and the pooled sample for the cohort source. The frailty trajectory membership for the pooled sample was the same as that in the individual cohorts. We conducted an exploratory analysis to examine the effect of frailty trajectory classification on protein expression in LonGenity using linear regression adjusted for age, sex, and OPUS-OPEL status. We compared three frail trajectories combined (given the small sample sizes) versus the relatively stable group as the reference. Bonferroni corrected $p$-values $<1.17 \times 10^{-5}$ (0.05/4265) were considered significant. Analyses were performed using SAS 9.4 (SAS Institute, Cary, NC).

## Results

### Frailty trajectories

Fig 1 shows four frailty trajectory classes in LonGenity over 12 years (median 7.6 visits): relatively stable (34.4%) as well as mild (36.1%), moderate (24.1%), and severely frail (5.4%). Fig 2 shows that four distinct latent classes representing frailty trajectories were also seen in EAS (median 5.6 visits): relatively stable (22.9%), mild (37.1%), moderate (27.9%), and severely frail (12.1%). EAS had higher moderate and severely frail membership (40.0% vs. 29.5%) than LonGenity; as may be expected from older age and higher comorbidities score (1.81 vs. 1.22) in EAS.

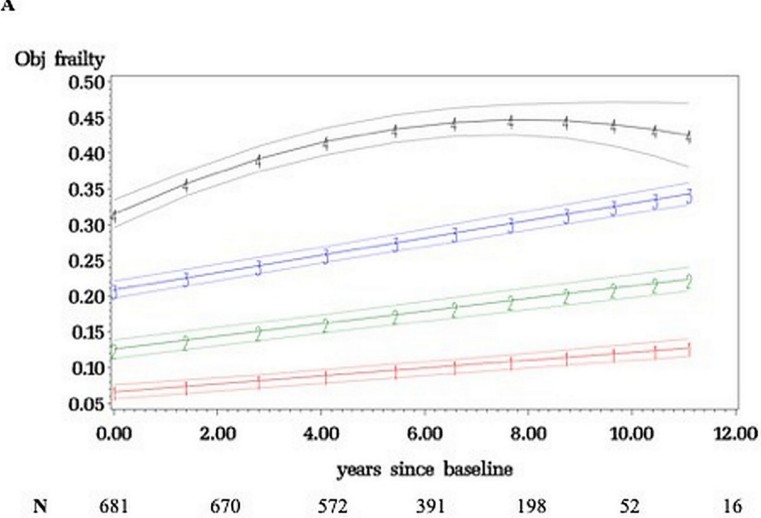

**Fig 1. Frailty trajectories with 95% confidence intervals in LonGenity cohort.** Four trajectory classes identified: relatively stable (1), mild frail (2), moderate frail (3) and severely frail (4). The Y-axis depicts the frailty index scores (range 0 to 1) and x-axis follow-up time in years.

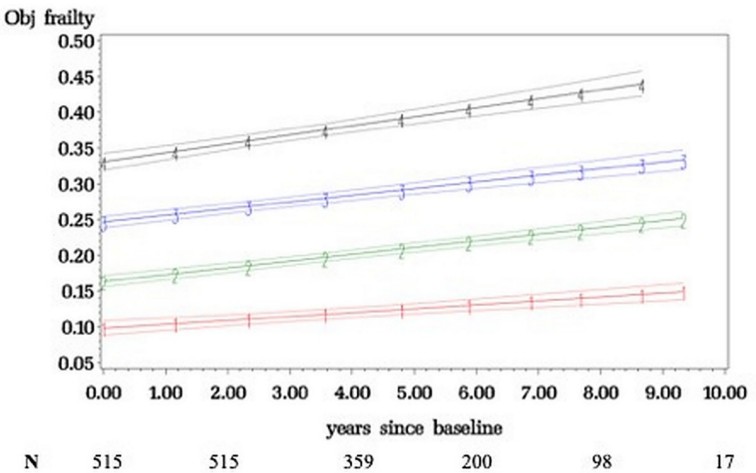

**Fig 2. Frailty trajectories with 95% confidence intervals in Einstein Aging Study cohort.** Four trajectory classes identified: relatively stable (1), mild frail (2), moderate frail (3) and severely frail (4). The Y-axis depicts the frailty index scores (range 0 to 1) and x-axis follow-up time in years.

Results in Figs 1 or 2 did not fundamentally change when age, sex, and OPEL-OPUS status were included as risk factors for class membership in the latent class trajectory analysis. Sample size is smaller in later years due to new recruits and attrition. The modeling utilizes the entire follow-up, but trajectories were similar even after restricting follow-up to six years in both cohorts.

Mean FI score increased from 0·06 at entry to 0·10 at final wave in the best trajectory, or 1·31 additional deficits accumulated on the average over follow-up. Mean FI score increased from 0·32 to 0·43 in the worst trajectory (3·95 additional deficits). Time to accumulate one additional deficit on the FI varied: relatively stable (median 31 months), mild (27 months), moderate (17 months), and severely frail (20 months). At baseline, only 17·0% of individuals met an established FI cutscore of 0·21 for frailty and 17·6% had phenotypic frailty [1, 4]. At the final wave, 35·7% had FI frailty and 21·0% had phenotypic frailty.

## Clinical profile

The most favorable demographic and health profiles were seen in the stable group (Table 2). Moderate and severely frail groups had lower education, more lived alone, OPEL under-represented, more physically inactive, and more comorbidities. WWT speed was slower across trajectories (p<0·001). Individuals in moderate and severely frail trajectories had higher odds of having WWT speed in the lowest tertile (<64·30 cm/s) than mild and stable (OR 1·63, 95% CI 1·03–2·59). This WWT cutscore had a sensitivity of 44·2% and specificity 70·7%. Lowering WWT cutscore improves sensitivity but decreases specificity, and vice versa.

## Mortality

There were 57 deaths over a median follow-up of 7·6 years (IQR 5·29–9·27) in LonGenity, and 82 deaths over a median 5·1 year follow-up (IQR 3·08–7·43) in EAS. Worse frailty trajectories were associated with higher mortality in both cohorts (Table 3).

In the pooled sample (n = 1196) adjusted for age, sex, education, and comorbidity score, those in mild (HR 2·33), moderate (HR 2·49), and severely frail trajectories (HR 5·28) had

**Table 2. LonGenity cohort baseline characteristics by frailty trajectory class.** P values for trend.

| Variables | Relatively stable (n = 224) | Mild frail (n = 263) | Moderate frail (n = 159) | Severely frail (n = 35) | P-value |
|---|---|---|---|---|---|
| Age, y | 71·34 ± 6·61 | 74·65 ± 5·80 | 78·22 ± 5·96 | 78·22 ± 6·40 | <0·001 |
| Women, % | 54 | 54 | 57 | 65 | 0·52 |
| Education, y | 18·25 ± 2·44 | 17·96 ± 2·73 | 17·30 ± 2·94 | 16·86 ± 2·83 | <0·001 |
| Live alone, % | 24 | 36 | 39 | 46 | 0·004 |
| Socioeconomic status (≥ 2 times the poverty level), % | 93 | 91 | 89 | 87 | 0·47 |
| OPEL status, % | 59 | 53 | 45 | 43 | 0·04 |
| Phenotypic frailty, %* | 7 | 19 | 24 | 41 | <0·001 |
| Physical inactivity, % | 9 | 18 | 40 | 71 | <0·001 |
| Walking speed, mean ± SD | 120·47 ± 17·36 (n = 183) | 113·49 ± 16·83 (n = 212) | 101·75 ± 16·64 (n = 129) | 88·00 ± 14·03 (n = 30) | <0·001 |
| Walking while talking speed, mean ± SD | 84·66 ± 27·60 (n = 183) | 77·22 ± 26·42 (n = 219) | 72·42 ± 24·15 (n = 124) | 61·34 ± 16·43 (n = 30) | <0·001 |
| Obesity, % | 14 | 22 | 34 | 40 | <0·001 |
| Comorbidities score, mean ± SD | 0·57 ± 0·69 | 1·31 ± 0·96 | 1·76 ± 1·10 | 2·43 ± 1·12 | <0·001 |
| Hypertension, % | 21 | 49 | 56 | 81 | <0·001 |
| Myocardial infarction, % | 2 | 5 | 12 | 12 | 0·001 |
| Heart failure, % | 0 | 0 | 3 | 1 | 0·01 |
| Depression, % | 11 | 22 | 25 | 47 | <0·001 |
| Chronic lung disease, % | 2 | 2 | 3 | 6 | 0·46 |
| Parkinson's disease, % | 0 | 1 | 1 | 6 | <0·001 |
| Stroke, % | 0 | 2 | 4 | 17 | <0·001 |
| Osteoarthritis, % | 21 | 42 | 59 | 61 | <0·001 |
| Cancer, % | 23 | 40 | 36 | 43 | <0·001 |
| FCSRT, free recall, mean ± SD | 34·96 ± 4·39 | 34·03 ± 4·63 | 32·19 ± 5·43 | 33·09 ± 5·01 | <0·001 |
| DSST, mean ± SD | 67·02 ± 13·98 | 61·20 ± 13·39 | 55·93 ± 12·40 | 51·49 ± 13·79 | <0·001 |

*Phenotypic frailty was diagnosed as per criteria proposed by Fried and colleagues.[1]

** Obesity was defined as body mass index (the weight in kilograms divided by the square of the height in meters) of 30 or higher.

OPEL: Offspring of parents with exceptional longevity; SD: standard deviation, FCSRT: free and cued selective reminding test; DSST: digit symbol substitution test.

higher mortality compared to the stable group (Table 3, Fig 3). To examine the incremental validity of frailty trajectories for mortality over established survival predictors,[25] we further adjusted the model for baseline FI, BMI, walking speed, free recall, and DSST scores. This subgroup analysis in 1005 individuals (120 deaths) with data available on all covariates showed unchanged associations of mild (HR 2.88, 95% CI 1.38–6.04), moderate (HR 2.62, 95% CI 1.08–6.36), and severely frail trajectories (HR 5.47, 95% CI 1.82–16.45) with mortality. Among the included covariates, only age and male sex were significantly associated with mortality (S2 Table in S1 File).

## Proteomics

Proteomic analysis was completed in 671 LonGenity participants: 220 stable (reference) and 260 mild, 156 moderate and 35 severely frail groups. Of the 4265 proteins assayed, 11 proteins were significant after adjustments for age, sex and OPEL/OPUS status as well as multiple comparison corrections (Fig 4, S3 Table in S1 File). Two fatty acid binding proteins (FABP) and leptin were over-expressed in combined frail groups. Proteins over-expressed in the stable group were neurocan core, voltage-dependent calcium channel subunit alpha-2/delta-3, delta

**Table 3. Frailty trajectories and mortality risk in LonGenity, Einstein Aging Study and pooled sample.**

| Trajectories | Deaths/ N | Hazard ratio (95% CI)* |
|---|---|---|
| **LonGenity (n = 681)** ** | | |
| Relatively stable | 4 / 224 | Reference |
| Mild frail | 20 / 263 | 3·52 (1·16–10·62) |
| Moderate frail | 23 / 159 | 5·20 (1·62–16·65) |
| Severely frail | 10 / 35 | 15·82 (4·18–59·88) |
| **Einstein Aging Study (n = 515)** | | |
| Relatively stable | 11 / 121 | Reference |
| Mild frail | 34 / 193 | 1·94 (0·96–3·89) |
| Moderate frail | 22 / 142 | 1·77 (0·82–3·81) |
| Severely frail | 15 / 59 | 3·45 (1·54–7·71) |
| **Pooled sample (n = 1196)** ** | | |
| Relatively stable | 15 / 345 | Reference |
| Mild frail | 54 / 456 | 2·33 (1·30–4·18) |
| Moderate frail | 45 / 301 | 2·49 (1·33–4·66) |
| Severely frail | 25 / 94 | 5·28 (2·68–10·41) |

* Hazard ratio with 95% confidence intervals from Cox regression analysis adjusted for age, sex, years of education and comorbidities score.

** The LonGenity sample analysis was in addition adjusted for OPEL-OPUS status. The pooled sample was in addition adjusted for cohort source.

and notch-like epidermal growth factor-related receptor, epidermal growth factor receptor, anthrax toxin receptor protein 2 (ANTR2), oligodendrocyte-myelin glycoprotein, contactin-1, and glypican-3. To gain insights into pathways that may link the 11 proteins to frailty, we further adjusted for education years, comorbidities score, BMI, physical inactivity, free recall score, and DSST score. In this post-hoc analysis, 3 (FABP-adipocytes, leptin, and ANTR2) out of the 11 proteins were no longer significant.

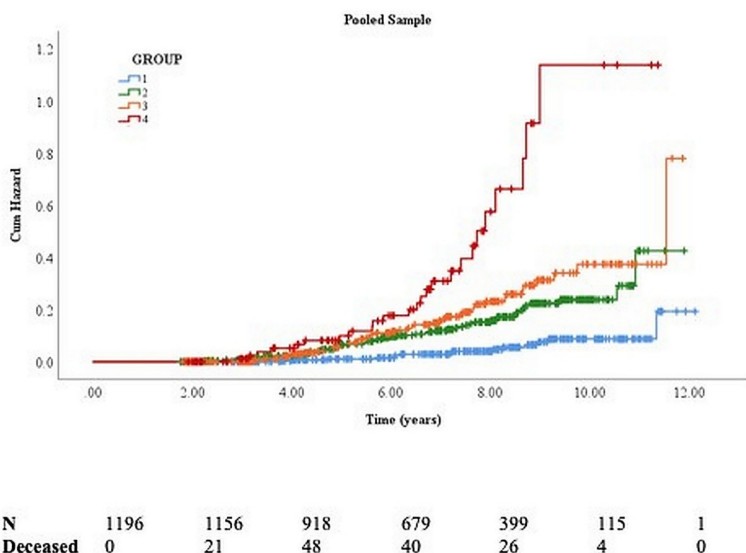

| N | 1196 | 1156 | 918 | 679 | 399 | 115 | 1 |
|---|---|---|---|---|---|---|---|
| Deceased | 0 | 21 | 48 | 40 | 26 | 4 | 0 |

**Fig 3. Survival plots show the cumulative risk of mortality based on frailty trajectories in the pooled LonGenity and Einstein Aging sample.** Group 1 (relatively stable–reference group), 2 (mild frail), 3 (moderate frail) and 4 (severely frail).

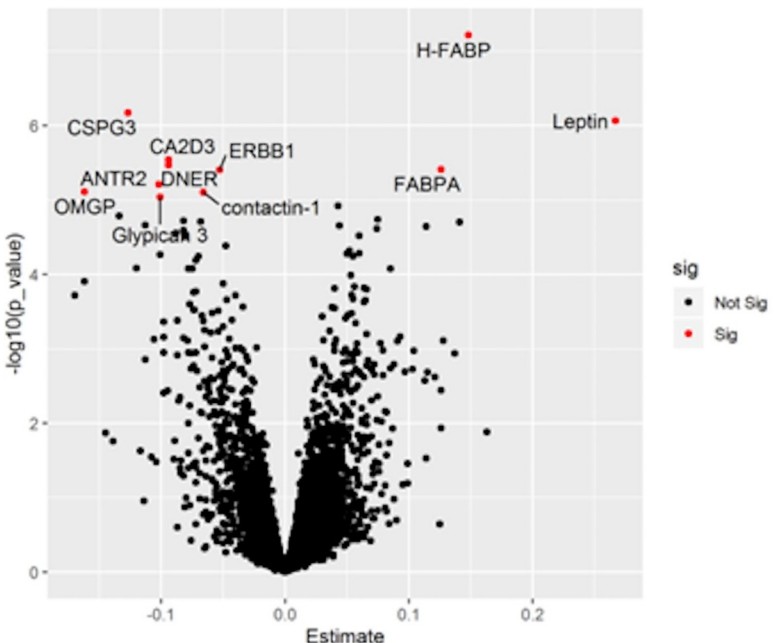

**Fig 4. Association of proteins with frailty trajectories.** Volcano plot with top proteins associated with frailty trajectories after Bonferroni correction annotated and in red (see S3 Table in S1 File). X-axis denotes Beta estimate coefficients from linear model. Y-axis shows significance levels presented as -$\log_{10}$ (*p*-value).

## Discussion

In this prospective cohort study of community-dwelling older adults, we identified four subgroups of individuals who differed in their trajectories of frailty from relatively stable to severely frail over 12 years. Four distinct classes of frailty trajectories were also shown in the racially and ethnically more heterogenous EAS cohort. The frailty trajectories were distinguished by clinical and proteomic signatures at baseline. Frailty measured over multiple timepoints was a more robust predictor of mortality compared to single timepoint measurement of frailty as well as other established survival predictors [25].

Individuals in the relatively stable frailty trajectory in the LonGenity cohort took 31 months to accumulate one additional deficit compared to only 20 months for the worst trajectory. Longitudinal studies show an increase in prevalence and incidence of frailty with advancing age [2, 5, 9, 26], which may be driven by those in worse frailty strata. Point-estimates of frailty do not adequately capture the variability in progression of frailty in populations [3]. There is a paucity of studies that use repeated-measures data to delineate subclasses of frailty trajectories that can help address this issue [3]. Frailty trajectories ranging from stable to severely frail were reported in few population-based studies using phenotypic frailty definitions [1, 10, 26]. In a study involving over 26,000 adults, age 75 and older, three frailty trajectories based on a 36-item FI were described with latent modelling over an one year period [11]. The majority of this sample was in a stable frailty trajectory group (76.6%), and the rest were in 'moderate growth' (21.2%), and 'rapidly rising' (2.2%) frailty trajectory groups [11]. Non-linear increases in overall frailty levels in other cohorts have been noted over longer follow-up periods [8]. Our study findings broadly replicate the different courses of frailty over a long period as well as in an independent cohort. But the proportion and rate of change in each class in the EAS and other cohorts can be different from that seen in the LonGenity study.

While higher baseline intercepts (frailer) were observed with worse frailty trajectories, only a minority of participants had frailty by phenotypic or FI criteria at entry or final visit. These baseline differences may reflect frailty accumulated over many years before enrollment. Current frailty assessments were developed for older populations [1, 2, 5]. Studies in younger populations may help delineate onset and course of frailty.

The progression of frailty can be influenced by risk as well as protective factors [8]. A number of potentially modifiable risk factors such as obesity, living alone, physical inactivity and comorbidities were related to the frailty trajectories in our study. This clinical profile can help guide strategies to promote health or treat frailty. The role of demographics such as age and sex, brain pathologies associated with dementia, and comorbid illnesses in the progression of frailty was highlighted in a recent review by Welstead and colleagues [8]. The higher proportion of OPEL in stable and mild classes suggests that frailty trajectories might in part be determined by longevity associated genetic factors [27]. We reported that OPEL had slower decline in physical function markers related to frailty compared to OPUS [6]. The WWT test [17] is a simple test that takes 1–2 minutes to administer, and predicts incident frailty and mortality [6, 17]. While WWT performance worsened across trajectories, its modest sensitivity does not lend itself to risk stratification in clinics. But it may be useful in research studies of risk and prognostic factors for frailty progression.

The clinical relevance of identifying frailty trajectories versus a single timepoint measure of frailty is supported by the higher mortality seen with worsening trajectories, even after accounting for baseline frailty level and several well-established survival predictors such as age, chronic illnesses and walking speed [25]. Despite the potential collinearity between baseline FI and frailty trajectories, only frailty trajectories predicted mortality in the fully adjusted model (see S2 Table in S1 File). Our findings are in line with Stow and colleagues who reported higher mortality in individuals in progressive frailty trajectory subclasses compared to individuals with relatively stable frailty trajectory membership [11]. Our findings do not imply that worse FI at baseline in individuals is inevitably linked to worse frailty trajectories over time. FI will show variability over time due to changes in underlying conditions as well as in response to treatments. Repeated frailty index measurement improved mortality prediction compared to single timepoint measurement in a population-based study leading the authors to recommend regular review of frailty status over follow-up [28].

Frailty and comorbid illnesses are distinct syndromes that show overlap [1, 3]. Progression of frailty and associated mortality risk can reflect worsening health or mobility. But mobility and chronic illnesses only accounted for 4·8% and 21·9% respectively of the 41 equally-weighted FI items. Moreover, mortality risk associated with frailty trajectories remained even after adjusting for walking speed and comorbidities [25]. Health and mobility are important contributors to frailty progression and mortality risk but other as yet unexplained factors may also have a role. Our results are supported by a few previous studies that described higher mortality in worse frailty trajectories defined using phenotypic [26] or FI approaches [9, 11].

Given the heterogeneous nature of the frailty syndrome, a single biomarker cannot efficiently predict or diagnose frailty. Identification of specific proteomic signatures as the first step toward a multi-marker approach is considered a very promising strategy in frailty diagnostics [29]. Recent exponential advances in proteomic technology enables a deeper investigation of biology. Proteins integrate environmental and genetic influences. As proteins are effectors of biological processes, they not only accurately predict pathology but also are potential therapeutic targets [30].

Our large-scale proteomic analysis linked three lipid metabolism related proteins to worse trajectories. Fatty acid-binding protein family are considered as lipid chaperones

[31]. The top hit, H-FABP, expressed mainly in heart and skeletal muscle, is involved in transport of long chain fatty acids from the cell membrane through the myocardial cytoplasm. H-FABP is a sensitive biomarker for cardiovascular risk factors [32]. FABP-Adipocytes (FABPA) is overexpressed with obesity as well as correlated with blood pressure and dyslipidemia [33]. The post hoc analysis adjusted for confounders including body composition and physical inactivity reduced the significance of leptin and FABPA, supporting a role for obesity in frailty progression. These findings are in line with previous studies that have linked lipid metabolism to aging and lifespan [14, 34]. Impaired adipose tissue function is reported to result in proinflammatory state as well as immune cell infiltration, senescent cells accumulation, and an increase in senescence-associated secretory phenotype, which can increase risk for frailty [34].

Chondroitin sulfate proteoglycan 3 (CSPG3), was the most downregulated protein in the combined frail classes. CSPG3 plays an important role in brain development [35]. CSPG3 and leptin were associated with prevalent frailty in LonGenity [13]. The gene (ANTXR2) coding the ANTR2 is linked to grip strength [36], a major frailty component [1]. Epidermal growth factor signaling may promote healthy aging,[37] and was negatively associated with frailty in LonGenity [13]. Adjustment for cognitive scores (to control for brain pathology) and other confounders did not appreciably change the associations of these proteins to frailty. There was overlap in the proteomic profile previously associated with frailty index at cross-section and trajectories in the LonGenity cohort [13]. Fatty acid-binding proteins and leptin were common in both frailty conditions. However, the growth factor pathway proteins were unique to the frailty trajectory suggesting common as well as distinct drivers of frailty status and future trajectories. The functionality of proteins associated with frailty trajectories need to be examined to follow up on our preliminary biological findings.

Limitations are noted. There is no gold standard for defining frailty though FI is widely used in research [2]. Both FI and phenotypic frailty definitions use a mix of subjective and objective items [1, 21]. Individuals might perceive subjective motoric changes before developing mobility disability [38]. Hence, subjective items might be sensitive to early changes in frailty compared to objective markers. LonGenity used parental longevity in an realtively homogenous Ashkenazi-Jewish sample as an entry criterion, which may limit generalizability, but biological discoveries from this cohort have been replicated in independent population samples [24, 27]. Besides, our findings regarding the number of frailty trajectories and their mortality risk was replicated in the ethnically and racially more heterogenous EAS cohort, which also had lower education and higher comorbidity levels. As the FI casts a broad net, a fully independent clinical marker was not available. We are unable to comment on onset or course of frailty in younger ages.

## Conclusions

Prevalence and incidence of frailty is high in older populations, and is recognized as an urgent public health topic to address [1, 3, 20]. But encouragingly, over half of individuals in our cohorts had stable or mild frailty over a long period, though the study samples were not selected to be representative of the general population. Public health initiatives to reduce frailty could, hence, focus resources on people in moderate or severe categories. FI has been calculated using automated data mining methods in large healthcare systems such as the National Health Service in the United Kingdom [11]. Automated approaches to derive frailty trajectories using commonly collected electronic medical data could be developed to implement our findings in clinical settings. The novel clinical and proteomic signals of frailty trajectories can help improve current risk assessments as well as point the way to new therapies.

## Supporting information

**S1 File.**
(PDF)

## Author Contributions

**Conceptualization:** Joe Verghese.

**Data curation:** Richard B. Lipton, Sofiya Milman, Nir Barzilai.

**Formal analysis:** Emmeline Ayers, Sanish Sathyan, Cuiling Wang.

**Funding acquisition:** Richard B. Lipton, Sofiya Milman, Nir Barzilai.

**Investigation:** Joe Verghese, Emmeline Ayers, Sanish Sathyan, Sofiya Milman.

**Methodology:** Sanish Sathyan, Richard B. Lipton, Sofiya Milman, Nir Barzilai, Cuiling Wang.

**Project administration:** Joe Verghese, Richard B. Lipton.

**Supervision:** Emmeline Ayers.

**Validation:** Cuiling Wang.

**Writing – original draft:** Joe Verghese.

**Writing – review & editing:** Emmeline Ayers, Sanish Sathyan, Richard B. Lipton, Sofiya Milman, Nir Barzilai, Cuiling Wang.

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
