## [Decision Letter · Decision Letter 0]

12 May 2021

PONE-D-21-08554

Trajectories Of Frailty In Aging

PLOS ONE

Dear Dr. Verghese,

Thank you for submitting your manuscript to PLOS ONE. After careful consideration, we feel that it has merit but does not fully meet PLOS ONE’s publication criteria as it currently stands. Therefore, we invite you to submit a revised version of the manuscript that addresses the points raised during the review process.

We look forward to receiving your revised manuscript.

Kind regards,

Antony Bayer

Academic Editor

PLOS ONE

Journal Requirements:

'Funding/Support: No funding received. The parent study funding is as follows. The LonGenity

cohort was funded by the National Institutes of Health grants P01AG021654, R01AG046949,

R01AG057909, R01AG044829, R01AG061155, RO1AGO57548, and K23AG051148, the

Nathan Shock Center of Excellence for the Biology of Aging P30AG038072, American

Federation for Aging Research, and Glenn Center for the Biology of Human Aging Paul Glenn

Foundation Grant. The Einstein Aging Study was supported by National Institutes of Health

grants P01AG03949, R01AG039409, RO1 AGO57548, and R21 AG056920. The sponsors of

the parent studies had no role in the design and conduct of the study; collection, management,

analysis, and interpretation of the data; and preparation, review, or approval of the manuscript.'

'The authors received no specific funding for this work. Parent study funding disclosed in manuscript.'

4. Please include a copy of Table 3 which you refer to in your text on page 25.

5. Please include captions for your Supporting Information files at the end of your manuscript, and update any in-text citations to match accordingly. Please see our Supporting Information guidelines for more information: http://journals.plos.org/plosone/s/supporting-information

Reviewers' comments:

Reviewer's Responses to Questions

**Comments to the Author**

1. Is the manuscript technically sound, and do the data support the conclusions?

Reviewer #1: No

2. Has the statistical analysis been performed appropriately and rigorously? 

Reviewer #1: No

3. Have the authors made all data underlying the findings in their manuscript fully available?

Reviewer #1: No

4. Is the manuscript presented in an intelligible fashion and written in standard English?

Reviewer #1: Yes

5. Review Comments to the Author

Reviewer #1: Thank you for inviting me to review this interesting paper on frailty trajectories. The authors present analyses of data from two US based cohorts: LonGenity and EAS. These longitudinal studies collect a range of information including health, sociodemographic and biological samples, allowing the authors to create a frailty index containing 41 items. The authors used latent growth mixture models to identify frailty trajectories in the two cohorts and found a four-class model to be the best fit in each. The trajectories characterised by higher intercepts (EAS) or higher intercepts and slopes (LonGenity) are associated with reduced survival probability. The authors find that these trajectories are also associated with sociodemographic factors, some of which may be amenable to intervention.

Overall this is a well written paper, which would contribute to the maturing literature on frailty and heterogeneous trajectories. I think the authors need to be a little more cautious about the novelty of their work (given the number of studies in this area) and the generalizability of the findings given the cohorts they have used (Regionally very specific, and in the case of LonGenity highly selected). I think readers would also benefit from more detail in the methods / results to aid interpretation of the trajectories and how they compare to other work.

Minor comments

TITLE

“Trajectories of frailty in Aging” – I might have missed a longer title (I know PLOS encourage full and truncated versions), but assuming this is the full title I think the article would be easier to discover if the title were more descriptive (study design, timescale etc - PLOS One guidelines are as follows “Specific, descriptive, concise, and comprehensible to readers outside the field, eg Impact of cigarette smoke exposure on innate immunity: A Caenorhabditis elegans model”)

ABSTRACT

(Background) I disagree that little is known – there’s been a lot of work in this area over the last few years (see linked review - 25 studies on trajectories https://academic.oup.com/gerontologist/advance-article/doi/10.1093/geront/gnaa061/5850544

It would be helpful if the authors could be clearer about the aim of the study here too.

(Methods) can you clarify here that the cohort are derived from the LonGenity study (i.e. 681 / ~60% of 1060 LonGenity participants)

(Methods) It would be helpful to describe the method used to derive the trajectories here

(Methods) Genetic/proteome work: I’m not clear why this was done - clarifying the aims could address this though.

(findings) some elements probably belong in methods, i.e. duplication in another cohort, the fact you pooled the cohorts for the mortality / survival analyses.

(findings) Is low education a modifiable risk factor in the cohort studied?

(Conclusion) I think some acknowledgement that half of the pooled sample is probably not representative of the general population should be made here.

INTRO

“The few previous studies using different definitions of frailty”

Per comment on the abstract , I think there are more than a few studies looking at frailty trajectories. I do agree with the authors that there is still much to be gained by thinking about which factors that influence trajectory membership

Elements of the second paragraph of the introduction duplicate the methods section (from “to address this knowledge gap” to “higher mortality risk”)

I think the clarity of the introduction and of the paper as a whole would be aided by the addition of a sentence or two clearly stating the aims of the study (same for the abstract)

METHOD

Readers would benefit from more information about the cohorts here – how long Is follow-up for EAS? For example

P5 – “no significant differences” : is this statistical or clinical significance? 1/3 of LonGenity and ½ of EAS were excluded from this study

P5 – “Phenotypic frailty” : I’m unclear as to why this is here (doesn't seem to have been used elsewhere?)

P6 - Was the duplication in the other cohort using pre specified start values? or was the model building process run entirelt independently?

RESULTS

P8

Readers would benefit from a little more information on the model selection / description. You fitted linear to cubic trends but which were selected for the final model? BIC and plausibility are stated in the methods, but how close was the BIC for 3 and 4 classes for example, and how did scientific plausibility guide the process (see comment on recovery in severe trajectory class in LonGenity)

Typo: calssess -> classes

“did not change after adjustment for age sex…”: can you give more detail? Were intercepts/slopes regressed on age / sex? How did you decide that the results did not change? Was it visual inspection of the plot or were model fit indices used to make this decision?

A table 1 containing demographic information for the 2 cohorts would be helpful here (current table 1, which contains analytic elements then becomes table 2)

Figure 1a – the most severe trajectory here seems to be the one that benefited from the quadratic component of the model and implies frailty improves: is improvement possible with the frailty measures you used, is this an artefact of differential survival, or is this improvement an artifact of the quadratic component (and if so I would query whether this was the right model to specify)

P9

Typo: suppementary -> supplementary

DISCUSSION

P10

“the four latent classes were replicated” – I’m not clear exactly what this means: is it that a 4 class models had the best fit in both cohorts (more information in results needed to support this) or is it that the 4 classes look the same in both cohorts, in which case from a purely visual interrogation I disagree (fig 1a/1b – EAS frailty profiles all have a higher baseline, severe trajectory in LonGenity has a clear quadratic arc, vs no curvature in EAS, slopes differ in Longenity, but the EAS trajectories do not vary much in the slopes / seem to differ in the intercept only)

P11

A few other studies are described here, but it’s not always clear what the link is to the authors own work – I think this section could be expanded to include references to the other studies on trajectories and give a direct comparison to this work (e.g. number of trajectory groups, proportion of people in each, and any associations with mortality/sociodemographic factors).

“Our study findings confirm…” – I think confirm is a little strong given the nature and size of the cohorts used here. These findings broadly replicate those from other studies looking at frailty trajectories, highlighting the potential for frailty measurement to aid prognostication, and demonstrating heterogeneity, often associated with sociodemographic factors (some of which may be amenable to intervention)

“Paucity of studies” – see above ref of review of longitudinal frailty studies from 2020

P14

“Besides, our findings regarding number of frailty trajectories… was replicated” – as above this might be the case but it isn’t clear from the results as currently described.

6. PLOS authors have the option to publish the peer review history of their article (what does this mean?). If published, this will include your full peer review and any attached files.

Reviewer #1: No

---

## [Author Response · Author response to Decision Letter 0]

10 Jun 2021

Editorial comments

Response: Thank you. We have reformatted the manuscript to comply with the journal format requirements, and will make any additional changes required.

2. We note that you have indicated that data from this study are available upon request. PLOS only allows data to be available upon request if there are legal or ethical restrictions on sharing data publicly. For information on unacceptable data access restrictions, please see http://journals.plos.org/plosone/s/data-availability#loc-unacceptable-data-access-restrictions. We will update your Data Availability statement on your behalf to reflect the information you provide.

Response: Thank you. Kindly update data availability section as follows: 

The Institutional Review Board of the Albert Einstein College of Medicine has data access restrictions in place because health claims data are a sensitive data source and have ethical restrictions imposed due to concerns regarding privacy. Anonymized data are available to all interested individuals or institutions upon request, and completion of a data transfer agreement. Please contact the Einstein Institutional Review Board staff at (irb@einsteinmed.org) for all data requests. 

3. Thank you for stating the following in the Acknowledgments Section of your manuscript: 'Funding/Support: No funding received. The parent study funding is as follows. The LonGenity cohort was funded by the National Institutes of Health grants P01AG021654, R01AG046949, R01AG057909, R01AG044829, R01AG061155, RO1AGO57548, and K23AG051148, the Nathan Shock Center of Excellence for the Biology of Aging P30AG038072, American Federation for Aging Research, and Glenn Center for the Biology of Human Aging Paul Glenn Foundation Grant. The Einstein Aging Study was supported by National Institutes of Health grants P01AG03949, R01AG039409, RO1 AGO57548, and R21 AG056920. The sponsors of the parent studies had no role in the design and conduct of the study; collection, management, analysis, and interpretation of the data; and preparation, review, or approval of the manuscript.'

Response: We have removed funding-related text from the manuscript. Kindly update Funding Statement of the online submission form as follows: 

‘The authors received no specific funding for this work. The parent study funding is as follows. The LonGenity cohort was funded by the National Institutes of Health grants P01AG021654, R01AG046949, R01AG057909, R01AG044829, R01AG061155, RO1AGO57548, and K23AG051148, the Nathan Shock Center of Excellence for the Biology of Aging P30AG038072, American Federation for Aging Research, and Glenn Center for the Biology of Human Aging Paul Glenn Foundation Grant. The Einstein Aging Study was supported by National Institutes of Health grants P01AG03949, R01AG039409, RO1 AGO57548, and R21 AG056920. The sponsors of the parent studies had no role in the design and conduct of the study; collection, management, analysis, and interpretation of the data; and preparation, review, or approval of the manuscript.' 

4. Please include a copy of Table 3 which you refer to in your text on page 25.

Response: ‘Table 3’ was included in the manuscript as ‘Supplementary Table 3.’ We have corrected the description in the text (Page 16) as well as in Figure 3 legend. 

5. Please include captions for your Supporting Information files at the end of your manuscript, and update any in-text citations to match accordingly. Please see our Supporting Information guidelines for more information: http://journals.plos.org/plosone/s/supporting-information

Response: Thank you. We have included captions for supplementary tables (Page 31), and updated in-text citations.

Reviewer 1

Thank you for inviting me to review this interesting paper on frailty trajectories. The authors present analyses of data from two US based cohorts: LonGenity and EAS. These longitudinal studies collect a range of information including health, sociodemographic and biological samples, allowing the authors to create a frailty index containing 41 items. The authors used latent growth mixture models to identify frailty trajectories in the two cohorts and found a four-class model to be the best fit in each. The trajectories characterised by higher intercepts (EAS) or higher intercepts and slopes (LonGenity) are associated with reduced survival probability. The authors find that these trajectories are also associated with sociodemographic factors, some of which may be amenable to intervention.

Overall this is a well written paper, which would contribute to the maturing literature on frailty and heterogeneous trajectories. I think the authors need to be a little more cautious about the novelty of their work (given the number of studies in this area) and the generalizability of the findings given the cohorts they have used (Regionally very specific, and in the case of LonGenity highly selected). I think readers would also benefit from more detail in the methods / results to aid interpretation of the trajectories and how they compare to other work.

Response: Thank you for the positive comments. We have followed the Reviewer’s suggestion for being cautious about the novelty throughout the manuscript, and noting limitations to generalizability. We thank the Reviewer for pointing us to a highly relevant recent review of frailty that we have incorporated into the manuscript (reference 8). We have added more details in Methods and Results as detailed in our responses to the comments below. 

Minor comments

1. TITLE: “Trajectories of frailty in Aging” – I might have missed a longer title (I know PLOS encourage full and truncated versions), but assuming this is the full title I think the article would be easier to discover if the title were more descriptive (study design, timescale etc - PLOS One guidelines are as follows “Specific, descriptive, concise, and comprehensible to readers outside the field, eg Impact of cigarette smoke exposure on innate immunity: A Caenorhabditis elegans model”)

Response: As suggested by the reviewer, we expanded the title as follows: “Trajectories of frailty in Aging: prospective cohort study.”

2. ABSTRACT: (Background) I disagree that little is known – there’s been a lot of work in this area over the last few years (see linked review - 25 studies on trajectories https://academic.oup.com/gerontologist/advance-article/doi/10.1093/geront/gnaa061/5850544 It would be helpful if the authors could be clearer about the aim of the study here too.

Response: We revised the background statement in the Abstract as follows: Emerging evidence suggests that there is significant variability in the progression of frailty in aging. We aimed to identify latent subpopulations of frailty trajectories, and examine their clinical and biological correlates. 

Thank you for pointing us towards this highly relevant review. We had already included individual studies from this review in our previous submission. But have now revised our manuscript based on this review (new reference 8).

3. (Methods) can you clarify here that the cohort are derived from the LonGenity study (i.e. 681 / ~60% of 1060 LonGenity participants)

Response: We included this clarification (Page 6). We also added a new Table 1 with cohort details as suggested by the reviewer.

4. (Methods) It would be helpful to describe the method used to derive the trajectories here

Response: As suggested, we moved the section on latent class modelling to the frailty section as well as expanded on the methods as requested by the reviewer (Page 8).

5. (Methods) Genetic/proteome work: I’m not clear why this was done - clarifying the aims could address this though.

Response: We revised the Introduction to more clearly state our biological aim (Page 6), and also noted the lack of biological studies of latent classes of frailty (Page 5). Our examination of proteomic signatures of trajectories follows our cross-sectional study of proteomics of frailty (reference 13, Introduction Page 5), and this biological investigation could spur research into protective mechanisms against frailty and help develop new interventions (Page 5, 20-21).

6. (findings) some elements probably belong in methods, i.e. duplication in another cohort, the fact you pooled the cohorts for the mortality / survival analyses.

Response: We have added or expanded on these elements in the Methods for latent modelling (Page 8) as well as mortality analysis (Page 10). 

7. (findings) Is low education a modifiable risk factor in the cohort studied?

Response: Education was not significant when examined in the mortality analysis model fully adjusted for all covariates, and the result was presented in supplementary Table 2.

8. (Conclusion) I think some acknowledgement that half of the pooled sample is probably not representative of the general population should be made here.

Response: We added this caveat (Page 22).

9. INTRO “The few previous studies using different definitions of frailty” Per comment on the abstract , I think there are more than a few studies looking at frailty trajectories. I do agree with the authors that there is still much to be gained by thinking about which factors that influence trajectory membership

Response: We agree, and have revised the Abstract, and Introduction (Page 5).

10. Elements of the second paragraph of the introduction duplicate the methods section (from “to address this knowledge gap” to “higher mortality risk”)

Response: We revised the text to reduce redundancy (Page 5-6)

11. I think the clarity of the introduction and of the paper as a whole would be aided by the addition of a sentence or two clearly stating the aims of the study (same for the abstract)

Response: Thank you for the suggestion. We revised the Introduction (Page 5-6) and Abstract to more clearly state our aims.

12. METHOD. Readers would benefit from more information about the cohorts here – how long Is follow-up for EAS? For example

Response: We have included the follow-up information for both cohorts as requested (Page 6). We also added new Table 1 as suggested by the reviewer to provide more cohort details.

13. P5 – “no significant differences” : is this statistical or clinical significance? 1/3 of LonGenity and ½ of EAS were excluded from this study

Response: We clarified that it was statistical difference on key variables (Page 6).

14. P5 – “Phenotypic frailty” : I’m unclear as to why this is here (doesn't seem to have been used elsewhere?)

Response: We had provided the phenotypic frailty definition as we had described the prevalence of phenotypic frailty in Results (Page 11) and in Table 2 as well discussed it with reference to frailty trajectories (Page 18).

15. P6 - Was the duplication in the other cohort using pre specified start values? or was the model building process run entirelt independently?

Response: We have clarified that the model was run independently in EAS (Page 10).

16. RESULTS. P8: Readers would benefit from a little more information on the model selection / description. You fitted linear to cubic trends but which were selected for the final model? BIC and plausibility are stated in the methods, but how close was the BIC for 3 and 4 classes for example, and how did scientific plausibility guide the process (see comment on recovery in severe trajectory class in LonGenity).

Response: We have expanded on the methodology of model selection as requested (Page 8). Given a fixed number of classes (denoted as K), we examined linear to cubic trends within each class by testing of the parameters and model fitting statistics (BIC), to obtain the final model given the number of classes, starting with k=1. To compare model fitting between K=k+1 versus k number of classes, we followed the recommendation based on change in BIC as described in Jones, Nagin and Roeder 2001 (reference 22), in the Methods section, to determine whether the model with k+1 classes is better than the model with k classes. For example, for the question about comparing 3 and 4 classes, the change of BIC from 3 to 4 classes models is 183, which suggests ‘strong’ evidence that 4-class model is better based on the cut-off laid out in Jones, Nagin and Roeder (2001). Judgement of plausibility, including how well the groups were seperated and the proportions of class membership, was also used in the determination of number of classes. 

17. Typo: calssess -> classes

Response: Thank you. Corrected (Page 10).

18. “did not change after adjustment for age sex…”: can you give more detail? Were intercepts/slopes regressed on age / sex? How did you decide that the results did not change? Was it visual inspection of the plot or were model fit indices used to make this decision?

Response: The result did not fundamentally change when age, sex, and OPEL-OPUS status were included as risk factors for class membership in the latent class trajectory analysis (revised Page 11).

19. A table 1 containing demographic information for the 2 cohorts would be helpful here (current table 1, which contains analytic elements then becomes table 2).

Response: Thank you for the suggestion. We added a new Table 1 with cohort details.

20. Figure 1a – the most severe trajectory here seems to be the one that benefited from the quadratic component of the model and implies frailty improves: is improvement possible with the frailty measures you used, is this an artefact of differential survival, or is this improvement an artifact of the quadratic component (and if so I would query whether this was the right model to specify)

Response: As indicated in the Methods section (Page 8), our model allows missing data under the assumption that the data are missing at random, i.e., the missing data process (due to survival, drop out, etc.) can depend on the observed outcome. The case of missing not at random is not testable without additional data. The improvement at later follow-up time shown in the most severe group is due to the quadratic trend. We chose this model because it is better than the linear model. Even though slight improvement is shown at later follow-up time due to the quadratic trend in this group, the dominating trend is still worsening over time in the group and overall worse than the other groups. If we fit a model with linear trend in all the 4 classes, overall similar group separation and pattern were observed with 86% overlap in group assignment. Note that the proportion in the most severe group is the lowest (5.4%, n=35) which shows evidence the presence of a small most severe group in the overall trajectory.

21. P9. Typo: suppementary -> supplementary

Response: Thank you. Corrected (Page 15).

22. DISCUSSION P10 “the four latent classes were replicated” – I’m not clear exactly what this means: is it that a 4 class models had the best fit in both cohorts (more information in results needed to support this) or is it that the 4 classes look the same in both cohorts, in which case from a purely visual interrogation I disagree (fig 1a/1b – EAS frailty profiles all have a higher baseline, severe trajectory in LonGenity has a clear quadratic arc, vs no curvature in EAS, slopes differ in Longenity, but the EAS trajectories do not vary much in the slopes / seem to differ in the intercept only)

Response: In the phrase “the four latent classes were replicated”, we meant that the existence of 4 latent classes was also shown in EAS data, but the pattern in each class can be different from that in Longevity study. The same method of latent class trajectory analysis as described in the method section was applied to EAS. We acknowledge that this can cause confusion, and therefore revised the text in the Abstract and Discussion (Page 18). 

23. P11. A few other studies are described here, but it’s not always clear what the link is to the authors own work – I think this section could be expanded to include references to the other studies on trajectories and give a direct comparison to this work (e.g. number of trajectory groups, proportion of people in each, and any associations with mortality/sociodemographic factors).

Response: We confirm that none of the studies referenced in the highlighted section were from our group (Page 19). There were only 2 studies in the review (reference 8) suggested by the reviewer that used mixture models to identify latent subpopulations of FI trajectories; similar to the aim of our study. The other studies in the same review reported non-linear trends in frailty progression without examining subclasses. The study by Stow et al (reference 11) was the only one to described latent subclasses in their overall sample. We had included this study in our submission but have now expanded its description (Page 19). The other study (Stephan et al, 2020) in the review reported latent classes of frailty by sociodemographic features, and not in the overall sample. As recommended, we have expanded the section (Page 19) to describe group membership in the Stow study, non-linear patterns in other studies as well as association with socio/demographic factors (Page 18). 

24. “Our study findings confirm…” – I think confirm is a little strong given the nature and size of the cohorts used here. These findings broadly replicate those from other studies looking at frailty trajectories, highlighting the potential for frailty measurement to aid prognostication, and demonstrating heterogeneity, often associated with sociodemographic factors (some of which may be amenable to intervention)

Response: We revised the terminology as recommended (Page 21). 

25. “Paucity of studies” – see above ref of review of longitudinal frailty studies from 2020

Response: Thanks again for pointing out this review. We have used this reference (8) to bolster our Introduction and Discussion. 

P14 “Besides, our findings regarding number of frailty trajectories… was replicated” – as above this might be the case but it isn’t clear from the results as currently described.

Response: We agree that the statement was not clear and have revised accordingly as described in response to comment 22 above.

---

## [Editor Report · Decision Letter 1]

17 Jun 2021

Trajectories of frailty in aging: Prospective cohort study

PONE-D-21-08554R1

Dear Dr. Verghese,

Thank you for your revised manuscript and thank you for your detailed attention to the reviewer's comments. I hope that you would agree that the paper has now been strengthened significantly. The only issue I noted is the affiliation of Dr. Wang - should this be "5" rather than "4"?

Consequently, we’re pleased to inform you that your manuscript has been judged scientifically suitable for publication and will be formally accepted for publication once it meets all outstanding technical requirements.

Kind regards,

Antony Bayer

Academic Editor

PLOS ONE
---

## [Editor Report · Acceptance letter]

1 Jul 2021

PONE-D-21-08554R1 

Trajectories of frailty in aging: Prospective cohort study 

Dear Dr. Verghese:

I'm pleased to inform you that your manuscript has been deemed suitable for publication in PLOS ONE. Congratulations! Your manuscript is now with our production department. 

Kind regards, 

on behalf of

Professor Antony Bayer 

Academic Editor

PLOS ONE